# Attitudes toward COVID-19 Vaccines among Patients with Complex Non-Communicable Disease and Their Caregivers in Rural Malawi

**DOI:** 10.3390/vaccines10050792

**Published:** 2022-05-17

**Authors:** Moses Banda Aron, Emilia Connolly, Kaylin Vrkljan, Haules Robbins Zaniku, Revelation Nyirongo, Bright Mailosi, Todd Ruderman, Dale A Barnhart

**Affiliations:** 1Partners In Health/Abwenzi Pa za Umoyo (PIH/APZU), Neno P.O. Box 56, Malawi; econolloy@pih.org (E.C.); rnyirongo@pih.org (R.N.); bmailosi@pih.org (B.M.); truderman@pih.org (T.R.); 2Division of Pediatrics, University of Cincinnati College of Medicine, 3230 Eden Ave, Cincinnati, OH 45267, USA; 3Division of Hospital Medicine, Cincinnati Children’s Hospital Medical Center, 3333 Burnet Ave, Cincinnati, OH 45529, USA; 4Harvard College, Harvard University, Cambridge, MA 02138, USA; kaylinvrkljan@college.harvard.edu; 5Ministry of Health, Neno District Health Office, Neno P.O. Box 52, Malawi; hrzanikuh@gmail.com; 6Partners In Health/Inshuti Mu Buzima (PIH/IMB), Kigali P.O. Box 3432, Rwanda; dbarnhart@pih.org; 7Department of Global Health and Social Medicine, Harvard Medical School, Boston, MA 02138, USA

**Keywords:** COVID-19, non-communicable diseases, vaccination hesitancy, COVID-19 vaccine, intention to vaccinate

## Abstract

Current low COVID-19 vaccination rates in low- and middle-income countries reflect an inequitable global vaccine distribution; however, local attitudes towards the COVID-19 vaccine are an important factor to meet vaccination benchmarks. We describe attitudes toward the uptake of the COVID-19 vaccine and perceptions among patients with NCDs and their caregivers using cross-sectional data collected through telephone interviews in Neno, Malawi. Out of 126 survey respondents, 71% were patients, and 29% were caregivers. Twenty-two percent of respondents had received at least one dose at the interview (95% CI: 15–30%), with 19% being fully vaccinated. Only 24% (95% CI: 12–40%) of unvaccinated respondents reported that they would accept an approved vaccine if it were offered today. Vaccines were perceived as unsafe or designed to harm and commonly associated with death, severe disability, infertility, and evil. However, over two-thirds reported high levels of trust in health care workers (73%) and community health workers (72%) as sources of information for the COVID-19 vaccine. Although the uptake of COVID-19 vaccine in this vulnerable population was three times than the national average, a low intention to be vaccinated persists among the unvaccinated. Strong trust in health care workers suggests that community engagement could help increase vaccine acceptance.

## 1. Introduction

Coronavirus disease (COVID-19), caused by the SARS-CoV-2 virus, was declared a pandemic on 11 March 2020 by the World Health Organization [1]. People of any age can suffer from COVID-19; however, older adults and people with comorbidities, especially heart disease, diabetes, and lung disease, have a higher mortality and morbidity due to COVID-19 [2]. Fortunately, people who receive a COVID-19 vaccine, including those with chronic conditions, have lower risk of hospitalization, disease severity, and death than unvaccinated individuals [3,4,5,6]. Safe and effective vaccines are one vital tool for controlling the COVID-19 pandemic; however, this impact can only be achieved with high vaccine coverage [7].

Inequitable practices in global vaccine manufacturing, purchasing, and distribution have resulted in poor vaccine access in low- and middle-income countries (LMICs), particularly in sub-Saharan Africa. These inequitable practices remain the primary barrier to adequate COVID-19 vaccination rates [8]. However, as vaccines become more accessible in LMICs, understanding local attitudes towards COVID-19 vaccines will become critical to ensure high vaccine acceptance.

There has been hesitancy against COVID-19 vaccines in both high- and low-income settings. Individuals often have negative perceptions concerning the vaccines’ safety and side effects, a low perceived risk of COVID-19, religious beliefs, mistrust in use, and efficacy [9,10]. Early in the pandemic—before COVID-19 vaccines were widely available—individuals in LMICs reported high vaccine acceptance [11,12]. However, vaccine hesitancy is a context-dependent phenomenon that changes over time, and as such, there are still significant gaps in our understanding of current attitudes towards the COVID-19 vaccine in LMICs.

Malawi declared a state of national disaster due to the COVID-19 pandemic on 20 March 2020, and registered its first confirmed coronavirus case on 2 April 2020 [13]. Before vaccine rollout in Malawi, studies estimated that 79% of Africans and 83% of Malawians were willing to be vaccinated [11,14]. On 5 March 2021, the country received its first 360,000 doses of the COVID-19 vaccines, which were prioritized for those with a higher risk of COVID-19 infection and complications, such as healthcare workers and individuals older than 60 years [15]. In April 2021, during early phases of the vaccination campaign, telephones surveys still found high initial levels of vaccine acceptance (12%) and intention to vaccinate (61%) in Malawi [16]. However, by the time members of the general population became eligible to receive vaccines in July 2021, a lack of funding and support for the local cold supply and storage chain, an insufficient workforce, and unreliable electricity and roads have led to some vaccines expiring before they could be used [17]. At this time, early reports of vaccine hesitancy were attributed to poor engagement with local authorities, the absence of coordinated information campaigns, and the spread of misinformation through social media [18]. As of 11 March 2022, Malawi had administered 1,955,495 million doses of COVID-19 vaccine and 4.4% of the total population had been fully vaccinated [19].

This evolution underscores the gaps in current knowledge about current attitudes towards COVID-19 vaccines in Malawi. Studies that were developed before COVID-19 vaccines were widely available in Malawi often asked about a hypothetical COVID-19 vaccine and do not necessarily reflect the population’s willingness to vaccinate after being exposed to specific rumors about the vaccine. Furthermore, none of these reports looked explicitly at vaccine hesitancy among rural populations in Malawi, who likely have less access to information distributed by the Ministry of Health and are potentially at higher risk of holding misinformation regarding vaccines, or among patients with non-communicable diseases, who are a high-priority group for vaccination due to their vulnerability to COVID-19. This study aimed to describe the COVID-19 vaccine uptake, intentions to vaccinate, reasons for and against vaccination, and sources of trusted information about COVID-19 vaccines among patients with complex NCDs and their caregivers in Neno District, Malawi.

## 2. Materials and Methods

### 2.1. Study Setting

We conducted the study in the Neno District located in the southwest part of Malawi, about 145 km from Blantyre, the country’s commercial city and 340 km from Lilongwe, its capital city. It is one of the most remote districts in the country, with no tarmac road leading to the district hospital. The district spans over 1561 square kilometers accounting for only 1.56% of Malawi’s total land area. Neno shares its boundaries with Ntcheu District to the North, Balaka and Zomba districts to the Northeast, Blantyre District to the east, Chikwawa to the South, Mwanza and People’s Republic of Mozambique to the south west and west respectively [20]. As of 2021, the district was home to an estimated 147,272 people [21]. Neno recorded its first COVID-19 case, a returnee from South Africa on 30 May 2020. As of 23 April 2022, the cumulative prevalence of COVID-19 in Neno was 7 per 1000 population with fatality of 25 per 1000 cases [22].

In 2007, Partners In Health (PIH), a United States-based organization working in eleven countries to provide a preferential option for the poor in health care, was invited to Neno District by Malawi’s Ministry of Health (MOH) to accompany them in strengthening health service delivery. Since 2015, PIH has partnered with the local facilities to provide NCD patients with care at each of the 14 health facilities in the district. In 2018 PEN-Plus clinics were established in Neno District Hospital and Lisungwi Community Hospital [23]. PEN-Plus is a WHO-supported model to improve preventative, outpatient services at first-level hospitals for complex NCDs, such as type 1 diabetes, sickle cell disease, and rheumatic heart disease [24]. Due to their pre-existing conditions, all PEN-Plus patients were eligible for the vaccine and vaccines were routinely available at the Neno and Lisungwi hospitals.

### 2.2. Study Design and Study Population

This cross-sectional study was nested in a prospective open cohort study among patients with complex NCDs in Neno. Starting in December 2020, all patients enrolled in the PEN-Plus clinic who had a telephone number on file were invited to participate in a prospective telephone cohort. Of the 450 patients enrolled in the PEN-Plus clinics in December 2020, 23% of the population (*n* = 105) had phone coverage. To maintain continuity of care during the COVID-19 pandemic, an additional fifty phones were distributed to patients increasing the coverage to 34% (*n* = 155). We conducted four rounds of telephone-based data collection. Questions related to COVID-19 vaccines were added to the survey in the fourth round of data collection from 27 September to 22 October 2021. In all rounds, caregivers responded on their behalf of patients who were less than 18 years of age or those who were very sick.

### 2.3. Data Collection

We recruited two enumerators to conduct interviews via telephone and entered the survey responses into a CommCare application. CommCare is an easily customizable, open source mobile platform where forms or survey questionnaires are programmed and allows for both online and offline data collection [25]. For each patient who had responded to one of the previous rounds of the telephone survey, enumerators made up to five attempts (one attempt per day) to contact participants by telephone. After five attempts, patients who could not be contacted were referred to the clinical team for an in-person visit to verify patient well-being. The survey questionnaire included demographic information of patients and caregivers and questions about the COVID-19 vaccine. We asked survey respondents to respond to the vaccination module from their perspective, whether they were the patient or the caregiver. Our vaccine questions were informed by a literature review and developed in consultation with the Partners In Health Cross-Site COVID-19 Cohort Research Network, a team of clinicians and researchers from eight PIH-supported countries and methodologists from Harvard Medical School and Partners In Health.

After identifying and reviewing questions from the relevant literature [9,12,14,26], the team prioritized key concepts during a group meeting. Following this meeting, one team member drafted an initial set of questions, which were then revised iteratively among team members until a final set of questions were agreed upon. This final set of recommended questions was reviewed by the study team in Malawi to ensure that it captured all locally relevant details and for translation in Chichewa. The resulting data collection tool was focused on two primary outcomes: “vaccine uptake”, which was defined as whether or not a person has received a COVID-19 vaccine and reflects both vaccine access and willingness to be vaccinated, and “intention to vaccinate”, defined as whether a person would receive an approved COVID-19 vaccine if it were made available to them for free today. When reporting on intention to vaccinate, we asked already vaccinated patients to imagine that they had never been vaccinated before. These two concepts were identified as distinct and important by our international team because we are often working in settings where logistic barriers, such as stockouts and transportation barriers, prevent individuals from accessing desired health services. Our team of experts additionally placed high priority on understanding patients’ reasons for vaccine uptake and intention to vaccinate as these reasons would provide actionable insights on how to design effective vaccination campaigns. Therefore, for patients who had refused, intended to refuse, or were unsure whether they would receive a COVID-19 vaccine, data collectors used open ended questions to probe for reasons they did not want to be vaccinated. By using open-ended questions, we aimed to understand individuals’ primary reasons against vaccination without priming them to respond to less important reasons.

Data collectors conducted real-time coding by categorizing participant responses using a pre-defined list of hypothesized reasons. Data collectors could code each response as corresponding to multiple pre-defined reasons and were encouraged to use a free-text “other” option to capture unexpected reasons against vaccination or additional specific details. We used a similar approach to identify reasons to want to be vaccinated among patients who had accepted, intended to accept, or were unsure whether they would accept a vaccine. We used open-ended questions to ask patients to report on their sources of information about the COVID-19 vaccine followed by closed-ended questions assessing their level of trust in various sources of information using a 5-point Likert scale that included “strongly distrust”, “distrust a little”, “neither trust nor distrust”, “trust a little”, and “trust a lot” options, as well as a “do not know” option. The vaccine module of our questionnaire is available in Appendix A. Due to an error in programming for one skip pattern, data were missing on intention to vaccinate among 57 unvaccinated individuals.

### 2.4. Data Analysis

Data were extracted from the CommCare database and analyzed in Stata version 15.1 [27]. We used descriptive statistics to describe and quantify close-ended questions, including the participant and study respondent characteristics. Point estimates and 95% confidence intervals for vaccine uptake were calculated for the overall study population and separately for subpopulations of patients and caregivers. Due to our skip-pattern error, point estimates and 95% confidence intervals for intention to vaccinate were reported separately for vaccinated and unvaccinated respondents. Open-ended data, specifically reasons for and against COVID-19 vaccination were analyzed by first reviewing the free-text responses and either including them into one of the existing pre-defined categories or creating a new category that reflected these responses. We then reported the percentage of respondents mentioning each reason and presented selected quotes from the open-ended responses to better contextualize participants’ views. We also reported the percentages of respondents using each potential information of information about the COVID-19 vaccine overall and by vaccination intention. We tested for differences in the source of information by vaccination intention status using a chi-squared test. For our analysis of respondents’ trust in each source of information, we combined responses of “neither trust nor distrust” and “don’t know” into a single neutral category and reported frequencies of respondents in each category.

## 3. Results

### 3.1. Demographic Characteristics, Vaccine Uptake, and Intention to Vaccinate

Out of 192 participants who had responded to one of the previous three surveys, 66% individuals responded to the fourth round of data collection (N = 126). Out of 126 respondents, 71% were patients themselves (*n* = 90) whereas the remainder were caregivers (*n* = 36, Table 1). Throughout this paper, N reflects the total number of respondents included in each analysis or sub-analysis while n reflects the number of individual with a particular trait or characteristic.

There were slightly more males than females. Approximately equal numbers of respondents were under 39 years of age (33%), 40–59 years of age (38%), and 60 years of age or older (29%). Overall, 59% of the respondents reported having had an opportunity to be vaccinated against COVID-19 before the survey. Only 22% of respondents had received at least one dose at the interview (95% CI: 15–30%). Most of the respondents who had received at least one dose were fully vaccinated (85%). Vaccine uptake was higher among patients (26%, 95% CI: 17–36%) than among caregivers (14%, 95% CI: 5–29%), but this difference was not statistically significant (*p* = 0.235). Among the 28 respondents who had already received the vaccine, only 82% would agree to be vaccinated again if they were offered the vaccine for the first time today (95% CI: 63–94%). For the 98 unvaccinated respondents, 57 were not asked to report vaccine intention due to a skip pattern error. For the remaining 41 unvaccinated respondents, only 24% (95%CI: 12–40%) would accept a vaccine if offered today, 76% (95%CI: 60–88%) would not, and no patient was unsure.

### 3.2. Reasons to Not Be Vaccinated

Figure 1a,b show that the two most common reasons mentioned for refusing the vaccine was that it was “unsafe” or “designed to harm”.

Patients perceived that the impacts of the vaccine on their health would be severe, including things like death, infertility, and disability:

“…Some people will die within a few years. Others will become disabled. Others will be so sick after the vaccine...”

“…The vaccine was meant to reduce the population of people in the world...”

Interestingly, many patients who had previously “refused” a vaccine cited other health problems (26%) as a reason for not being vaccinated. Participants cited health care workers as playing a role in their refusal.

“…Those with non-communicable diseases, they will get very sick if they receive the vaccine….”

“…She is expectant, the clinician advised not to get the jab….”

Other common reasons not to want a vaccine included distrust in the specific vaccine type and religious beliefs.

“…People are saying that there are special vaccines for the medical team and the local people….”

“…The vaccine is a symbol of 666, which is associated with Satanism….”

Others wondered about the usefulness of the vaccine, as stated by one NCD patient:

“…People who were vaccinated are also getting sick, therefore the vaccine is useless.”

### 3.3. Reasons to Accept the COVID-19 Vaccine

Among the 33 respondents that would accept vaccination if it were offered today, the most commonly reported reasons were for their own health (88%), to stop the pandemic (73%), and for the health of their families (61%) (Figure 2).

Participants identified health care workers as trusted sources of information.

“…I was very impressed with the message from health officers that [for] patients with NCDs the immune system is deficient. Once they get infected with the disease, they will be critically ill and possibly lose their life….”

Some participants who had initially refused the vaccine reported reconsidering vaccination after watching friends receive the vaccine without side effects.

“…She has been impressed by people who got the jab. They are okay. No side effects as some people have been speculating….”

Individuals also mentioned the role that health workers and local leaders could play in helping people change minds.

“…there is a need for the health personnel to come to them with a strong message about the vaccine….”

“…he is not feeling any side effects after getting the jab as speculated by other people. He is the village headman; many people in his village are willing to be vaccinated...”

### 3.4. Sources of Information about the COVID-19 Vaccine

Among the total of 126 respondents, the most common source of information was mass media (74%), followed by health care workers (HCW) (53%), Government/Ministry of Health (MoH) (48%), and the least common being social media (3%) and their employer (2%). There was no significant difference in the source of information by vaccination intention (Figure 3).

Respondents reported having trust in facility-based health care workers (73%), the Ministry of Health (72%), and community health workers (72%) (Figure 4). In contrast, respondents reported having distrust for social media (38%), family and friends (25%), and local leaders (21%).

## 4. Discussion

Overall, our study of patients with complex NCDs and their caregivers living in rural Malawi found that vaccine uptake (22%) and intention to vaccinate among the unvaccinated (24%) were substantially lower than COVID-19 vaccine acceptance rates from a systematic review of African studies (49%) [28]. Intention to vaccinate was also lower than previously studies done in Malawi, which were completed before the COVID-19 vaccine was widely available [14,16]. Our respondents commonly perceived the vaccine as unsafe or designed to harm and commonly associated the vaccine with death, severe disability, infertility, and evil. Similar reasons to oppose COVID-19 vaccines have been reported elsewhere [14,28,29,30]. The lower vaccine acceptance in our study population could reflect changes in vaccines’ perceptions over time with actual vaccine campaigns. For example, specific rumors have been more common following global vaccine distribution. Similar findings have been reported in Hong Kong, where concerns about vaccine safety increased throughout the pandemic [31]. The lower vaccine acceptance could also reflect the fact that our study population is from a rural area. A study conducted in 10 countries across Asia, Africa, and South America found that people living in rural areas perceive new vaccines as riskier than people living in urban areas [32]. Higher levels of COVID-19 vaccine hesitancy were also reported among rural populations in Norway, India, the United States, and Bangladesh [33,34,35,36], although a few studies have found higher hesitancy urban compared to rural areas [33,37,38]. Our findings underscore that vaccine hesitancy is a dynamic and context-dependent challenge, and local ongoing monitoring and research are needed to monitor and address new challenges.

Although intention to vaccinate in our population was low compared to Malawi’s national average, our survey respondents were almost three times more likely to have one dose of the COVID-19 vaccine (22% vs. 7.9%) and over four times more likely to be fully vaccinated (19% vs. 4.4%) [19]. The NCD patients and their caregivers who participated in this survey would have likely had better access to vaccines and more opportunities to be educated about vaccines due to their long-standing relationship with and frequent visits to local district hospitals. They also may have been more motivated to be vaccinated because either they or their loved one had an NCD and were at elevated risk of serious illness from COVID-19. The patients and caregivers in our study trusted their healthcare workers to provide information about COVID-19 vaccines. Similarly, high levels of trust in health care workers have been observed in diverse settings, including the United States, Jordan, and other countries [12,39,40]. Many patients also cited their vaccinated neighbors or family members as influencing their willingness to be vaccinated. Peers’ acceptance of vaccines has previously been associated with increased intention to vaccinate against HPV and influenza and may have a similar relationship with the COVID-19 vaccine [41,42,43]. These finding suggest that testimony from health workers and community members can help promote COVID-19 vaccination.

As a result of this study, our clinical and community teams have adopted several new communication strategies to promote COVID-19 vaccination. First, we are educating healthcare workers to be vaccine ambassadors for patients. This process includes coaching healthcare workers to emphasize the importance of the vaccine to vulnerable patients, including those with chronic conditions such as NCDs and pregnant patients, during routine clinic visits. Furthermore, health care workers were encouraged to be honest with patients about potential side effects of the vaccine and the chance of breakthrough infections since patients are aware of these possibilities and minimizing these realities could lead to further distrust. Second, to dispel myths about COVID-19 vaccines, we are directly engaging community leaders, including religious leaders, village headmen, and local politicians, to provide vaccine education and discuss prevalent myths. After these education sessions, we are inviting these community members to share testimonies during a local radio program that will be broadcast in our district. Further research is required to ascertain if these interventions and processes and other strategies are improving COVID-19 uptake.

Our study has several limitations in generalizability. This cross-sectional telephone survey study was done among a select group of patients in rural Malawi and consequently our study population is not representative of all populations of Malawi or even within the Neno District. The results cannot necessarily be generalized to urban places within Malawi or to rural residents without mobile phones. However, the selection bias due to phone ownership was reduced by the PEN-Plus clinic’s mobile phone distribution program, which was designed to increase phone coverage among the poorest patients. Similarly, based on their long-standing relationship with the PEN-Plus clinic, we expect our respondents to have better access to and acceptance of vaccines than other rural Malawian populations. We would expect both of these limitations to result in our study underestimating the true prevalence of vaccine hesitancy among the general population in Neno. Additionally, our small sample size led to large standard errors and wide 95% confidence intervals, particularly for our vaccination intention questions, which had greater missingness due to a skip pattern error. Despite these limitations, we do believe that our findings demonstrate that vaccine hesitancy is a challenge in this region. Furthermore, we believe that the questions we used to assess vaccine uptake and intention to vaccinate, which were developed in a collaborative process that involved researchers from a team of 8 low-, middle-, and upper-income countries and are available in Appendix A, may be of interest to others researcher working in settings where low vaccine uptake reflects a combination of poor access to vaccines and poor attitudes towards vaccines. Finally, the inclusion of open-ended questions in our survey is a strength of this study because it allowed us to understand the reasons for vaccine hesitancy and acceptance and develop a communication strategy tailored to these reasons.

## 5. Conclusions

Vaccine uptake and intention to vaccinate among this group of vulnerable non-communicable disease patients and their caregivers in rural Malawi was three times higher than the general public’s current vaccination rate but substantially below the vaccination target of 70%. Respondents perceived the vaccine as unsafe and designed to harm and commonly associated the vaccine with death, severe disability, infertility, and evil. However, trust in health care workers suggests that further engagement could help increase vaccine acceptance.

## Figures and Tables

**Figure 1 vaccines-10-00792-f001:**
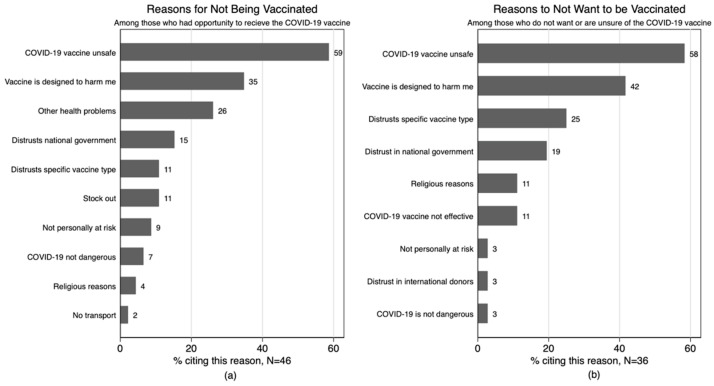
(**a**) Reasons for not being vaccinated among those who had the opportunity to receive the COVID-19 vaccine; (**b**) Reasons for not accepting a vaccine today. N reflects the total sample size for each panel.

**Figure 2 vaccines-10-00792-f002:**
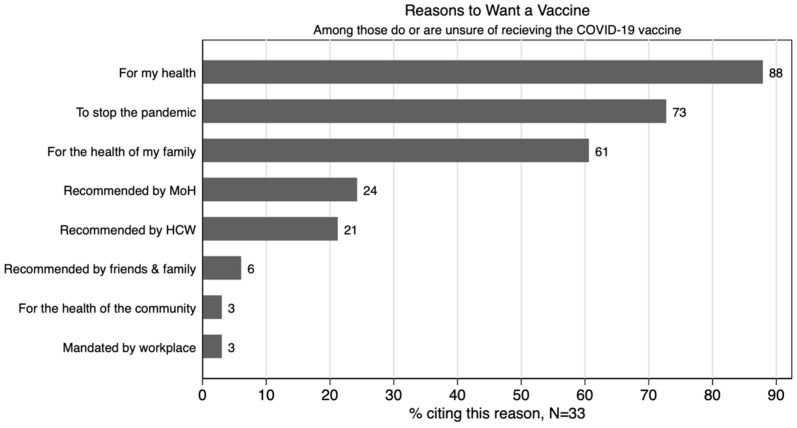
Reason to be vaccinated.

**Figure 3 vaccines-10-00792-f003:**
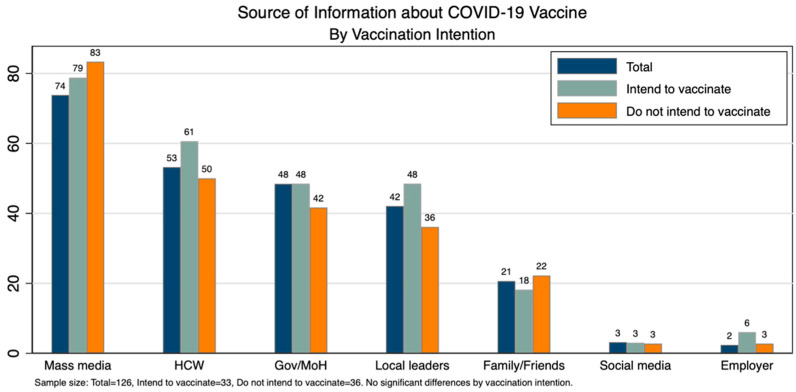
Source of information about COVID-19 Vaccine.

**Figure 4 vaccines-10-00792-f004:**
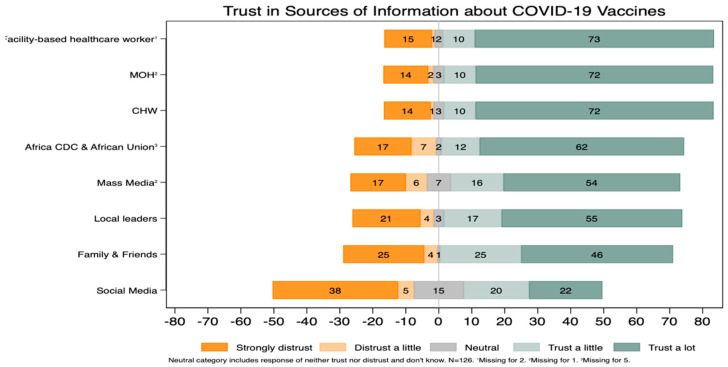
Trust in sources of information about COVID-19 by vaccine.

**Table 1 vaccines-10-00792-t001:** Characteristics of survey respondents and outcomes (*N* = 126).

	*n*	%
Respondent type		
Patient	90	71%
Caregiver	36	29%
Sex (*N* = 120)		
Male	40	44%
Female	50	56%
Age category (*N* = 120)		
18–39	40	33%
40–59	45	38%
≥60	35	29%
Vaccination Status (*N* = 126)		
Refused vaccine	46	37%
Partially vaccinated	4	3%
Fully vaccinated	24	19%
Had not had the opportunity to be vaccinated	52	41%
Intention to vaccinate among those with at least one dose (*N* = 28)		
Would accept a vaccine today	23	82%
Would not accept a vaccine today	5	18%
Unsure	0	0%
Intention to vaccinate among unvaccinated (*N* = 41) ^1^		
Would accept a vaccine today	10	24%
Would not accept a vaccine today	31	76%
Unsure	0	0%

^1^ Responses missing for 57 participants due to a skip pattern error.

## Data Availability

The data presented in this study are available on request from the corresponding author. The data are not publicly available due to privacy and confidentiality.

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
