# Peer review of "Attitudes toward COVID-19 Vaccines among Patients with Complex Non-Communicable Disease and Their Caregivers in Rural Malawi"

_vaccines, 2022, doi:10.3390/vaccines10050792_

Round 1

Reviewer 1 Report

The manuscript, which fits within the scope of this Journal, piqued my curiosity. The topic, in my opinion, is intriguing enough to pique the readers' interest. The main contribution fills the gaps that would open up challenging, interesting, and significant research directions.

To further their claims, the authors cited relevant literature and the survey fully clarifies its position in the current literature in this domain. 
 Overall the manuscript is just a compiled view of relevant theories that have been used to support the discussion as the authors' specified methodology using the right theory for the work.

The methodology adopted for this work is appropriate. 
The design is appropriate to fulfill the author´s research hypotheses. But I think that can be improved, mainly the efforts about the quality and quantity research. 

The structure employed for the categorization of these methods and techniques appears to require further refinement.

The results are presented clearly and analyzed appropriately and the conclusions are adequately tied together with other elements of the paper

The main contribution fills the gaps that would open up challenging, interesting, and significant research directions. The structure employed for the categorization of these methods and techniques appears to require further refinement.
There is a very nice attempt to discuss the advantages and limitations of the current techniques which are often presented in vague terms and with well-structured references to the specifics.

The article is interesting  and may provide important  information, is well written, well organized and provides the necessary background 
for the authors' approach. 

Reviewer 2 Report

This is a well-written and interesting paper about attitudes toward COVID-19 vaccines in Malawi.

Could you please add some information about the Neno district such as number of inhabitants. The paper would also be strengthened if you report some information about incidence rates of covid-19 as well as mortality.

CommCare -> please explain

I disagree in the authors claim that confidence intervals are less imprecise due to low sample size. They will be wider, but the confidence (and precision) will still be the same (95%).

The biggest concern is the sample size in the study. Could you elaborate a bit about representativity?

Do you have any idea about validity and reliability of the questions included in the survey?
